# Performance Evaluation and Application Field Analysis of Precise Point Positioning Based on Different Real-Time Augmentation Information

**Mengjun Wu [1], Le Wang [1],\*, Wei Xie [2] , Fan Yue [1] and Bobin Cui [1]**

1 School of Geological Engineering and Surveying, Chang'an University, Xi'an 710054, China; 2021126025@chd.edu.cn (M.W.); south_wind@chd.edu.cn (F.Y.); bobin@chd.edu.cn (B.C.)
2 National Time Service Center, Chinese Academy of Sciences, Xi'an 710600, China; xiewei@ntsc.ac.cn
\* Correspondence: wangle18@chd.edu.cn; Tel.: +86-134-7465-1614

**Abstract:** The most commonly used real-time augmentation services in China are the International GNSS Service's (IGS) real-time service (RTS), PPP-B2b service, and Double-Frequency Multi-Constellation (DFMC) service of the BeiDou Satellite-Based Augmentation System (BDSBAS) service. However, research on the performance evaluation, comparison, and application scope of these three products is still incomplete. This article introduces methods for obtaining real-time augmentation information and real-time orbit and clock offset recovery. Based on real-time orbit and clock offset accuracy, positioning accuracy, and positioning availability, this article systematically evaluates the performance and analyzes the application fields of Centre National d'Études Spatiales (CNES), PPP-B2b, and BDSBAS augmentation information. The results of the evaluation revealed that the radial accuracy of the CNES and PPP-B2b real-time orbit product is consistent, and the Root Mean Square (RMS) is better than 5 cm. The CNES real-time orbit product can achieve centimeter-level accuracy in both along-track and cross-track components, surpassing PPP-B2b's decimeter-level accuracy. Both services demonstrate consistent accuracy in the real-time clock offset, with PPP-B2b showing similar standard deviations (STDs) of 0.16 ns for different satellites. However, for CNES, the STD of the real-time clock offset varies, with values of 0.10 ns, 0.19 ns, and 0.60 ns, respectively, for GPS, BDS-3 Medium Earth Orbit (MEO), and BDS-3 Inclined Geosynchronous Satellite Orbit (IGSO) satellites. Centimeter-level accuracy is achieved after convergence and positioning availability exceeds 99% for CNES and PPP-B2b services. Therefore, the difference between the two services in application areas depends on the acquisition of augmentation information. However, BDSBAS, which concentrates on code observations, demonstrates inferior performance in real-time orbit, clock offset, positioning accuracy, and positioning availability when compared to the other two services. Its primary application is in the aviation and maritime domains, where there is a greater need for service integrity, continuity, and reliability.

**Keywords:** real-time augmentation; multifaceted evaluation; PPP; PPP-B2b; BDSBAS DFMC; application field

## 1. Introduction

On 1 April 2013, the International GNSS Service (IGS) launched the IGS Real-Time Service (RTS), which provides real-time augmentation information, including real-time orbits and clock offset corrections for real-time precise point positioning (RT PPP) services [1]. Out of all the IGS Analysis Centers, the real-time augmentation information from the Centre National d'Études Spatiales (CNES) Analysis Center now exhibits the highest accuracy [2]. The RTS depends on internet infrastructure to deliver real-time augmentation information. Therefore, the application and development of the RTS is restricted by the stability of internet infrastructure [3]. Additionally, since the official opening of the BeiDou Navigation Satellite System (BDS-3) on 31 July 2020, it has been developed rapidly and

widely used. The PPP-B2b service is a crucial element of BDS-3 and operates independently of internet infrastructure. PPP-B2b augmentation information, which currently supports only BDS-3 and GPS, will be broadcast in real time via Geostationary Earth Orbit (GEO) satellites for customers in China and surrounding regions [4]. In addition, BDS-3 has the BeiDou Satellite-Based Augmentation System (BDSBAS), which makes use of GEO satellites to give customers in China and the surrounding areas RT code correction data. It offers services to users in civil aviation, maritime, and other industries that demand high positioning accuracy, integrity, availability, and continuity, improving the accuracy and integrity of the basic satellite navigation system [5,6].

RT PPP is an important branch of PPP that has wider application scenarios and greater value than post-PPP. Therefore, many scholars have researched the theory, methods, and applications of RT PPP. Wang et al. [7,8] compared and analyzed the real-time products of several analysis centers, and came to the unanimous conclusion that CNES had the best precision products, with the best Root Mean Square (RMS) of the GPS orbit in each direction. Liu et al. [2,9] evaluated the performance of RTS products and came to the consistent conclusion that the standard deviation (STD) of the clock offset was less than 0.1 ns. The accuracy of GPS-only RT PPP in the east (E), north (N), and up (U) direction is about centimeter-level, with the vertical direction accuracy being marginally worse than the horizontal direction. When compared with the GPS-only solution, the performance of GPS/BDS-3 and GPS/BDS-2′s combined positioning accuracy was slightly improved.

Numerous academics have examined and assessed the PPP-B2b service in recent years. Lu et al. [10] conducted evaluation research on PPP-B2b real-time products and analyzed their time-varying characteristics and integrity. Song et al. [11–13] used a PPP-B2b real-time service with various frequency combinations to assess the PPP positioning performance. Based on B1I/B3I and B1C/B2a frequency combinations, they came to the conclusion that the positioning performance could satisfy the needs of the dynamic decimeter-level service in China. Xu and Tao et al. [13,14] compared the accuracy of PPP-B2b and CNES products in a multifaceted way. The analysis of the clock offset product's accuracy of BDS-3 revealed differences between the two. Tao et al. [14–16] evaluated the performance of the GPS/BDS-3 combination RT PPP based on the PPP-B2b service. Their research indicated that the STD of the RT clock offset is around 0.05~0.18 ns, the radial accuracy of the RT orbit is better than 10 cm, and the BDS-3/GPS dual system static RT PPP can achieve centimeter-level positioning. Guo et al. [17–19] assessed the service availability and PPP performance of the PPP-B2b service. Their research indicated that the availability of PPP-B2b exceeds 70% in Asia and over 80% in China. Compared to the broadcast ephemeris, the satellite clock offset and orbit accuracy can be improved. The static PPP accuracy reaches the centimeter level, while decimeter-level positioning performance can be achieved for kinematic PPP. Tang et al. [20] evaluated the performance of the BDS-3/GPS RT PPP time transfer by the PPP-B2b service, showing that sub-nanosecond-level accuracy can be achieved by the RT PPP time transfer by PPP-B2b, and the time transfer performance decreased as the elevation angle mask increased. Geng et al. [21] conducted a kinematic RT PPP positioning experiment based on the PPP-B2b service in the ocean. The ocean RT PPP corrected by PPP-B2b resulted in an RMS of 21.3 cm for BDS-3 only and 18.2 cm for GPS/BDS-3. The research demonstrates the value of PPP-B2b services for marine applications.

The BDSBAS plays a crucial role in BDS-3. Several scholars have evaluated and studied the BDSBAS dual-frequency augmentation service. For instance, Niu et al. [22,23] have examined GNSS's integrity and augmentation theory and methods, analyzed the key technology of decimeter augmentation technology, and assessed its accuracy. Zhang et al. [24,25] evaluated the availability of the BDSBAS single-frequency (SF) service, the results indicated its low availability, particularly at the boundary of the service area. Cao et al. [26,27] analyzed the performance of the Double-Frequency Multi-Constellation (DFMC) service of the BDSBAS in a multifaceted way. They came to the conclusion that dual-frequency augmentation has significantly improved positioning accuracy. The BDSBAS SF service's

positioning accuracy was around 1.0 m in the horizontal direction and 2.0 m in the vertical direction (95%). For the DFMC service, it was around 0.6 m and 1.2 m (95%), respectively.

As indicated by the above, there is no multifaceted comparison between the BDS-BAS, PPPB2b, and IGS nor any relevant elaboration on their applicable industries. This contribution focuses on the comparison among CNES, PPP-B2b, and BDSBAS real-time augmentation information. In Section 2, this paper will introduce the acquisition method of the real-time augmentation information and the recovery method of the real-time orbit and clock offset. It will comprehensively compare and analyze the three products, evaluating their orbit and clock offset accuracy. In Section 3, PPP's performance and availability are analyzed. Section 4 analyzes the application field. Finally, Section 5 presents this paper's conclusions about the performance evaluation of the three services.

## 2. Real-Time Augmentation Information Analyses

### *2.1. Real-Time Augmentation Information Acquisition*

(1)    IGS real-time augmentation information.

Via the Ntrip communication protocol, real-time augmentation information released by IGS analysis centers can be obtained. Users can apply for a Caster account on the IGS official website https://igs.org/rts/user-access (accessed on 6 June 2023) to obtain RTS data. In this contribution, RT augmentation information from CNES is used [3].

(2)    PPP-B2b real-time augmentation information.

PPP-B2b real-time augmentation information is transmitted through BDS-3 GEO satellites. Users can access PPP-B2b real-time services by using special receivers through B2b signals; then, by decoding the PPP-B2b signals, the real-time corrections can be obtained [4].

(3)    BDSBAS real-time augmentation information.

BDSBAS real-time service products are also broadcast through BDS-3 GEO satellites, which can be received by users through special receivers, that can decode B1C/B2a signals [5,6].

Unlike RTS services that rely on network transmission, the PPP-B2b and BDSBAS service products are limited to China and surrounding areas.

### *2.2. Real-Time Orbit and Clock Offset Recovery Method*

2.2.1. Real-Time Orbit Recovery

(1)    IGS real-time orbit recovery

When recovering IGS's real-time orbit, the Issue of Data (IOD) of the orbit correction should be matched with that of the broadcast ephemeris. For GPS satellites, its IOD can be obtained directly from the broadcast ephemeris, while the IOD of BDS should be calculated as follows [28,29]:

$$IODE_c = FMOD\left(\frac{toes}{720,240}\right) \tag{1}$$

where *IODEc* is the IOD calculated by BDS, *toes* is the BDS clock offset reference time, which can be obtained from the BDS broadcast ephemeris, and *FMOD* is the remainder function.

Assuming that *t*0 is the broadcasting time of the orbit correction message, in radial (R), along-track (A) and cross-track (C) directions, the satellites' position and velocity corrections are $(\delta_{t0}^R, \delta_{t0}^A, \delta_{t0}^C, \delta_{t0}'^R, \delta_{t0}'^A, \delta_{t0}'^C)$, and *t* is the time of the signal transmit in the satellite. At time *t*, the corrections of the satellite position are $\delta^R$, $\delta^A$, and $\delta^C$, respectively, which can be expressed as follows:

$$\begin{cases} \delta^R = \delta_{t0}^R + \delta_{t0}'^R(t - t0) \\ \delta^A = \delta_{t0}^A + \delta_{t0}'^A(t - t0) \\ \delta^C = \delta_{t0}^C + \delta_{t0}'^C(t - t0) \end{cases} \tag{2}$$

The satellite orbit corrections are converted from the satellite-fixed system to the Earth-Centered Earth-Fixed (ECEF) system, which can be expressed as follows:

$$\begin{bmatrix} \delta^X \\ \delta^Y \\ \delta^Z \end{bmatrix} = \begin{bmatrix} \vec{e^R} & \vec{e^A} & \vec{e^C} \end{bmatrix} \begin{bmatrix} \delta^R \\ \delta^A \\ \delta^C \end{bmatrix} \tag{3}$$

Under ECEF, the corrections to satellite positions in the *X*, *Y*, and *Z* directions are represented by $\delta^X$, $\delta^Y$, and $\delta^Z$, correspondingly. The unit vector in the R, A, and C directions is represented by $\vec{e^R}$, $\vec{e^A}$, and $\vec{e^C}$, respectively.

(2)　PPP-B2b real-time orbit recovery

For PPP-B2b real-time orbit recovery, multiple IODs should be correctly matched, including the IOD of state-space representation (SSR) corrections (IODSSR), the IOD of the satellite PRN mask (IODP), the IOD of the orbit and clock offset correction (IODCorr), and the IOD of navigation (IODN). The detailed matching rules for PPP B2b products are as follows in Table 1.

**Table 1.** Detailed matching rules for PPP-B2b products [4].

| IOD Type | Message Type |
| --- | --- |
| IODSSR | 1, 2, 3, and 4 |
| IODP | 1 and 4 |
| IODCorr | 2 and 4 |
| IODN | 2 and broadcast ephemeris |

Note that the correction of PPP-B2b is based on the CNAV1 broadcast ephemeris, which is broadcasted by the B1C signal. Compared to the BDS-2 broadcast ephemeris, CNAV1 includes 18 parameters; the difference is the calculation method of the long semi-axis and satellite motion speed [30].

Since there are only satellite position corrections in the three components, i.e., *R*, *A*, and *C*, at time *t*, the satellite position corrections are $\delta^R$, $\delta^A$, and $\delta^C$, respectively, which can be obtained from the following equations:

$$\begin{cases} \delta^R = \delta^R_{t0} \\ \delta^A = \delta^A_{t0} \\ \delta^C = \delta^C_{t0} \end{cases} \tag{4}$$

Then, transform the coordinate data according to Equation (3).

(3)　BDSBAS real-time orbit recovery

Because the BDSBAS orbit correction coordinate relies on the BeiDou Coordinate System (BDCS) [6], the corrections in the *X*, *Y*, and *Z* directions are $\delta^X$, $\delta^Y$, and $\delta^Z$, correspondingly, which can be obtained by Formula (5).

$$\begin{bmatrix} \delta^X \\ \delta^Y \\ \delta^Z \end{bmatrix} = \begin{bmatrix} \delta^X_{t0} \\ \delta^Y_{t0} \\ \delta^Z_{t0} \end{bmatrix} + \begin{bmatrix} \delta'^X_{t0} \\ \delta'^Y_{t0} \\ \delta'^Z_{t0} \end{bmatrix} * (t - t0) \tag{5}$$

Assuming that *t0* is the broadcasting time of the orbit correction message, in X, Y, and Z directions, the satellites' position and velocity corrections are ($\delta^X_{t0}$, $\delta^Y_{t0}$, $\delta^Z_{t0}$, $\delta'^X_{t0}$, $\delta'^Y_{t0}$, $\delta'^Z_{t0}$) in the BDCS.

Note that the correction of BDSBAS DFMC is based on the CNAV2 broadcast ephemeris, which is broadcasted by the B2a signal [6].

2.2.2. Real-Time Clock Offset Recovery

(1)    IGS real-time clock offset recovery

At time $t0$, the IOD of the clock offset correction should be matched with that of the broadcast ephemeris, then coefficients $C_0$, $C_1$, and $C_2$ are used to correct the IGS real-time clock offset. At time $t$, the correction for the satellite clock offset is represented by $\Delta C$, which can be calculated by Formula (6) [28,29]:

$$\Delta C = C_0 + C_1(t - t0) + C_2(t - t0)^2 \tag{6}$$

According to the principle of closest time and IOD matching, the matched broadcast ephemeris satellite clock offset is $dt0$, and the corrected clock offset $dt$, at time $t$, can be calculated by Equation (7).

$$dt = dt0 + \frac{\Delta C}{c} \tag{7}$$

where $c$ is the speed of light.

(2)    PPP-B2b real-time clock offset recovery

The only coefficient $C_0$ in the PPP-B2b clock offset correction, which is a negative value, can be determined by accurately matching the IOD as specified in Table 1. Thus, the satellite clock offset $dt$ at time $t$ can be determined using the following equation:

$$dt = dt0 - \frac{C_0}{c} \tag{8}$$

(3)    BDSBAS real-time clock offset recovery

At time $t$, the coefficients $C_0$ and $C_1$ are used to correct the BDSBAS RT clock offset $dt$, which can be calculated by Equation (9).

$$dt = dt0 + \frac{C_0 + C_1(t - t0)}{c} \tag{9}$$

Based on the above theoretical basis, this paper implements the recovery and performance evaluation of the real-time precision orbit and clock offset product of CENS, PPP-B2b, and BDSBAS DFMC services based on self-written software version 1.0.

*2.3. Real-Time Product Quality Analysis*

2.3.1. Real-Time Orbit Accuracy Analysis

To evaluate the accuracy of CNES, PPP-B2b, and BDSBAS orbit corrections, this paper compared the RT orbit, which was recovered by augmentation information, with the final orbit product from Wuhan University Multi-GNSS (WUM). The satellite orbit accuracy is evaluated using R, A, and C from DOY091 to DOY096, 2023, and then the RMS value was calculated.

(1)    GPS orbit accuracy

Figure 1 illustrates the average RMS values for the GPS satellite orbit accuracy of CNES, PPP-B2b, and BDSBAS over a span of 6 days. The data indicates that the GPS satellite orbit accuracy in the R direction from CNES is superior to 3.5 cm and surpasses the accuracy in the C and A components. Among the three directions, the A component exhibits the poorest accuracy. The average accuracy of the GPS RT orbit from CNES, in the R, A, and C directions, is 2.77 cm, 4.30 cm, and 3.72 cm, correspondingly. The RT orbit accuracy of PPP-B2b GPS satellites is poorer than CNES, especially for the A and C directions, which is 3.53 cm, 11.22 cm, and 14.85 cm in the R, A, and C directions, respectively. This is because the PPP-B2b real-time augmentation information is estimated using a limited number of Chinese regional stations, resulting in a restricted number of observable GPS satellites [3]. Furthermore, the BDSBAS relies on code correction, resulting in the worst

orbit accuracy among the three corrections. The accuracies for the R, A, and C direction are 19.20 cm, 62.70 cm, and 57.08 cm, respectively. During the Day Of Year (DOY) 091-096, only 18 GPS satellites corrections can be broadcast by BDSBAS, which is fewer than the number corrected by PPP-B2b (26) and CNES (32). This limitation is due to the BDSBAS service correction based on L1/L5 frequencies, and some GPS satellites do not support the L1/L5 frequencies [6].

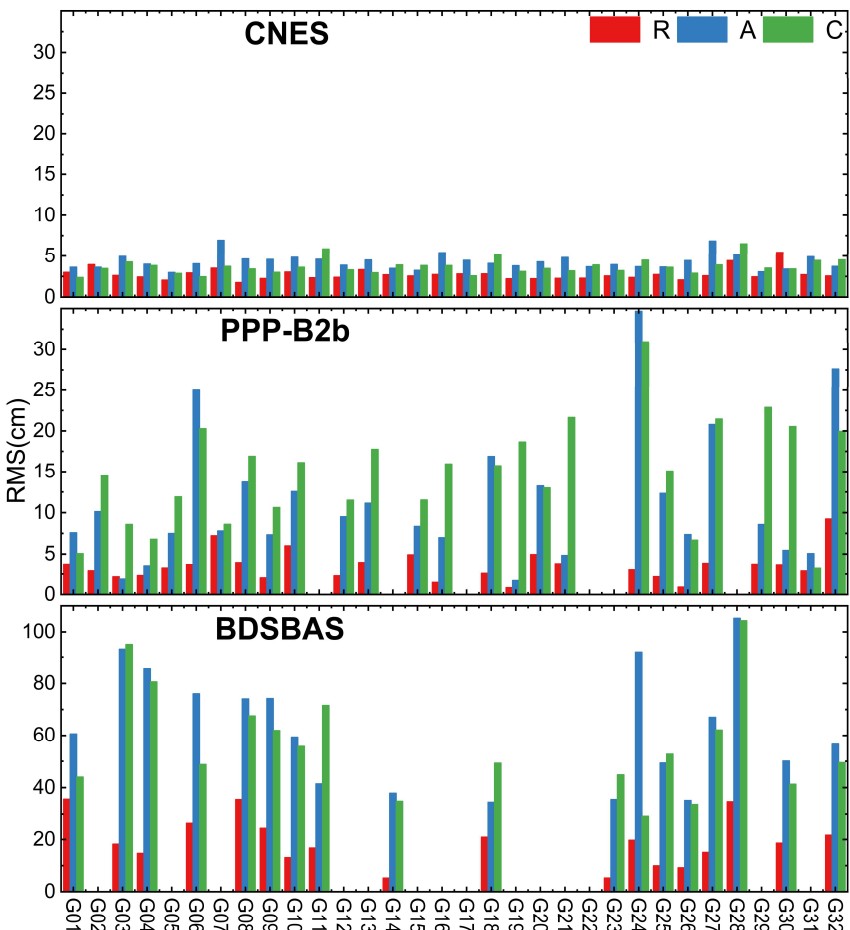

**Figure 1.** Average RMS of GPS satellite's orbit accuracy.

(2)    BDS-3 orbit accuracy

Figure 2 displays the average RMS of the RT orbit for BDS-3 Medium Earth Orbit (MEO) satellites from CNES, PPP-B2b, and BDSBAS over a period of 6 days. The average orbit accuracy in the R, A, and C component of BDS-3 MEO satellites for CNES are 4.59 cm, 7.06 cm, and 6.39 cm, respectively. The accuracy of the A and C component is marginally poorer than the R component for PPP-B2b, which is at the centimeter level and almost always superior to 5 cm, which is marginally greater than that of CNES, while the A and C direction are not. The mean RMS of it is 2.97 cm, 10.05 cm, and 11.11 cm, respectively. The RT augmentation information of the BDSBAS is based on code correction, resulting in a considerably lower orbit accuracy of BDSBAS BDS-3 MEO satellites than CNES and PPP-B2b, which is 11.09 cm, 37.70 cm, and 31.65 cm in the R, A, and C direction, respectively.

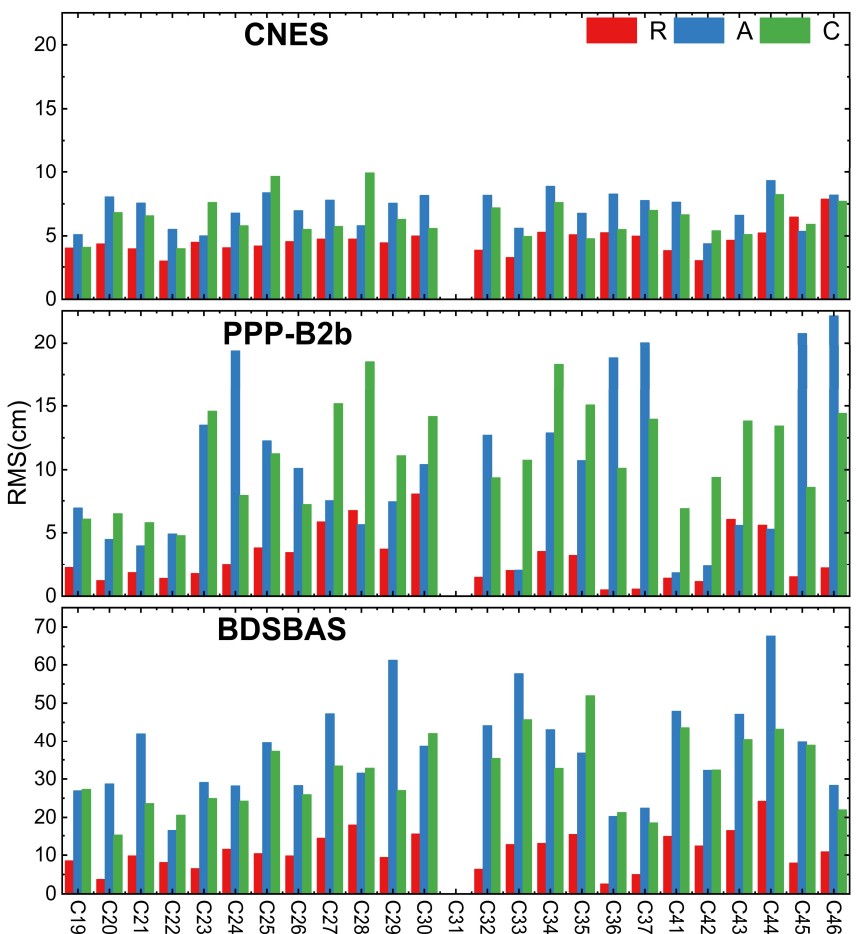

**Figure 2.** Average RMS of BDS-3 MEO satellite's orbit accuracy.

The average RMS values for the BDS-3 Inclined Geosynchronous Satellite Orbit (IGSO) satellites' orbit of CNES, PPP-B2b, and BDSBAS during a 6-day period are displayed in Figure 3. For the R, A, and C component, the accuracy of the BDS-3 IGSO satellites' orbit is 18.69 cm, 12.72 cm, and 12.42 cm for CNES and 6.75 cm, 22.52 cm, and 19.92 cm for PPP-B2b, correspondingly. The orbit accuracy of BDS-3 IGSO satellites based on BDSBAS augmentation information is significantly poorer than CNES and PPP-B2b since it is dependent on code observations, which, in the R, A, and C direction, is 9.11 cm, 47.93 cm, and 45.34 cm, respectively.

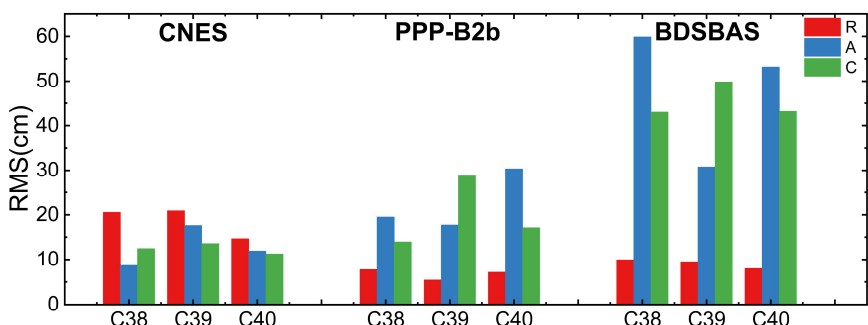

**Figure 3.** Average RMS of BDS-3 IGSO satellite's orbit accuracy.

In conclusion, for PPP-B2b, the BDS-3 satellites' orbit accuracy is marginally superior to CNES in the R direction, particularly for the IGSO satellites. Nevertheless, it is inferior to CNES's orbit accuracy in the A and C directions, particularly for GPS satellites. Further-

more, PPP-B2b satellites' orbit accuracy is decimeter-accurate in the A and C directions, which is significantly poorer than CNES, which could be as a result of there being fewer stations than CNES. CNES satellites' orbit accuracy is capable of achieving centimeter-level precision in all directions, with the exception of BDS-3 IGSO satellites. Based mostly on code rectification, the corrected orbit accuracy of the BDSBAS service is at the decimeter level, which is inferior to CNES and PPP-B2b, and the BDS-3 satellites' orbit accuracy is marginally higher than GPS.

### 2.3.2. Real-Time Clock Offset Accuracy Analysis

To assess the accuracy of the RT clock offset from CNES, PPP-B2b, and BDSBAS, this paper compared it with the final clock offset products, which are provided by WUM. To eliminate the impact of the different data on the clock offset performance evaluation, this paper chose the double-difference method. Finally, the STD of the double-difference clock offset series is calculated and used as the index for evaluating the clock offset. For CNES, the virtual reference clock offset method is used to eliminate the influence of the clock offset reference [28]. However, for PPP-B2b and BDSBAS products, this method cannot be adopted due to their discontinuous and inconsistent satellites for different epochs. G09 and C39 with longer arc segments are selected as reference satellites to eliminate the influence of the clock offset reference from DOY 092-096; as there is no C39 data in DOY 091 final clock offset products, C40 is used instead [29].

(1)  GPS clock offset accuracy

The STDs of the RT clock offset during a 6-day period for GPS satellites from CNES, PPP-B2b, and BDSBAS are displayed in Figure 4. For CNES, the STD of the various GPS satellites' RT clock offset is comparable, which is 0.10 ns. Nevertheless, there are fewer clock offsets of GPS satellite correction messages accessible as a result of PPP-B2b's signal load constraint. The STD of the GPS clock offset from PPP-B2b is 0.16 ns, and it is noticeably worse than CNES. The BDSBAS augmentation information is estimated based on code observations, resulting in a clock offset accuracy that is significantly worse than CNES and PPP-B2b, which is 1.63 ns.

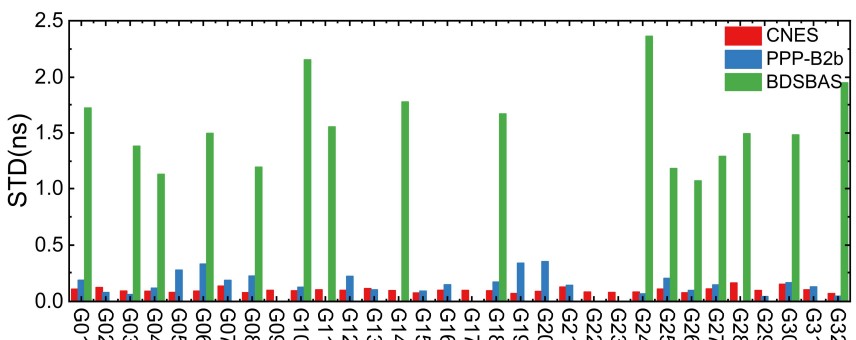

**Figure 4.** Average STD of GPS satellite's clock offset accuracy.

(2)  BDS-3 clock offset accuracy

The STD of the clock offset for CNES, PPP-B2b, and BDSBAS over a period of six days is shown in Figure 5. For CNES, BDS-3 MEO satellites have a clock offset accuracy of 0.19 ns, which is far greater than that of IGSO satellites, while the STD of IGSO satellites' clock offset is 0.60 ns. For PPP-B2b, with an STD of roughly 0.16 ns, the clock offset accuracy of the BDS-3 MEO and IGSO satellites exhibits little variation. In general, PPP-B2b BDS-3 satellites have a little higher clock offset accuracy than CNES. BDSBAS's clock offset is based on code observations, which makes it substantially less accurate than CNES and PPP-B2b. Its mean STD for the BDS-3 clock offset is 0.68 ns as well.

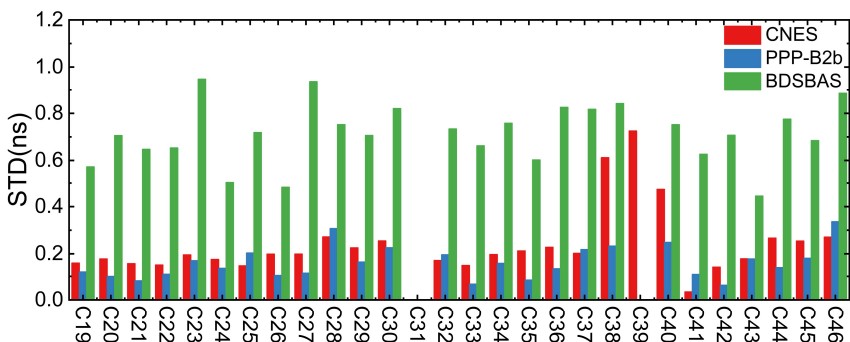

**Figure 5.** Average STD of BDS-3 satellite's clock offset accuracy.

The aforementioned study leads to the conclusion that CNES GPS satellites have better real-time clock offset accuracy than PPP-B2b. However, the BDS-3 satellites' clock offset accuracy from CNES is worse than that of PPP-B2b, particularly for IGSO satellites. Additionally, the clock offset accuracy of BDSBAS is the poorest among the three clock offset products, but its real-time clock offset accuracy reaches a sub-nanosecond level.

## 3. Real-Time PPP Performance Evaluation

### 3.1. Positioning Accuracy Analysis Based on Real-Time Augmentation Information

3.1.1. RT PPP Theory and Processing Strategy

The basic GNSS observation equations for RT PPP are as follows:

$$\begin{cases} P_{r,j}^s = \rho_r^s + c(dt_r - dt^s) + T_r^s + I_r^s + d_{r,j} - d_j^s + \varepsilon_{r,j,P}^s \\ L_{r,j}^s = \rho_r^s + c(dt_r - dt^s) + T_r^s - I_r^s + \lambda_j\left(N_{r,j}^s + b_{r,j} - b_j^s\right) + \varepsilon_{r,j,L}^s \end{cases} \tag{10}$$

where $P_{r,j}^s$ and $L_{r,j}^s$ represent pseudorange and carrier phase observations, respectively. $s$ and $r$ represent the satellite and receiver, respectively. $j$ represents the $j$th frequency. $\rho_r^s$ represents the distance between the phase centers of the satellite and receiver. $c$ represents the speed of light. $dt_r$ and $dt^s$ represent the receiver and satellite clock offsets, respectively. $T_r^s$ represents the slant tropospheric delay. $I_r^s$ represents the slant ionospheric delay. $d_{r,j}$ and $d_j^s$ represent pseudorange hardware delays for the receiver and satellite, respectively. $b_{r,j}$ and $b_j^s$ represent phase hardware delays at the $j$th frequency for the receiver and satellite. $\lambda_j$ represents the wavelength of the phase observation at the $j$th frequency. $\varepsilon_{r,j,P}^s$ and $\varepsilon_{r,j,L}^s$ represent the pseudorange and carrier phase observation noise.

The RT PPP function model employed in Section 3.1.2 is the ionosphere-free (IF) combination model, which refers to the elimination of the first-order ionospheric delay of pseudorange and carrier observations through linear combination. The dual-frequency IF combination observation can be expressed as follows:

$$\begin{cases} P_{r,IF}^S = \alpha_{1,2}P_{r,1}^S + \beta_{1,2}P_{r,2}^S \\ L_{r,IF}^S = \alpha_{1,2}L_{r,1}^S + \beta_{1,2}L_{r,2}^S \end{cases} \tag{11}$$

where $\alpha_{1,2} = \frac{f_1^2}{f_1^2 - f_2^2}$, $\beta_{1,2} = -\frac{f_2^2}{f_1^2 - f_2^2}$, $f_1$, and $f_2$ represent the frequencies of the two signals participating in the IF combination. By combining Formulas (10) and (11), we can form the IF combination model as follows:

$$\begin{cases} P_{r,IF}^S = \rho_r^s + c(dt_r - dt^s) + T_r + (\alpha_{1,2}d_{r,1} + \beta_{1,2}d_{r,2}) - (\alpha_{1,2}d_1^s + \beta_{1,2}d_2^s) + \varepsilon_{r,IF,P}^S \\ L_{r,IF}^S = \rho_r^s + c(dt_r - dt^s) + T_{rs} + (\alpha_{1,2}b_{r,1} + \beta_{1,2}b_{r,2}) - (\alpha_{1,2}b_1^s + \beta_{1,2}b_2^s) + \lambda_{IF}N_{r,IF}^s + \varepsilon_{r,IF,L}^S \end{cases} \tag{12}$$

where $\lambda_{IF}$ and $N^s_{r,IF}$ represent the wavelength and float ambiguity of the IF combination, respectively, and are expressed as follows:

$$\lambda_{IF} N^s_{r,IF} = \alpha_{1,2}\lambda_1 N^s_{r,1} + \beta_{1,2}\lambda_2 N^s_{r,2} \tag{13}$$

In the PPP processing process, the observation equation directly incorporates the precision clock offset as a true value, thereby eliminating the satellite clock offset. However, the current satellite clock offset products are all obtained using a dual-frequency IF combination method, which results in the clock offset products including the IF code delay. Therefore, the satellite clock offset products differ from the true clock offset. The relationship between the satellite clock offset $dt^s_{IF}$ product and the true clock offset $dt^s$ is expressed as follows:

$$dt^s_{IF} = dt^s + (\alpha_{1,2} d^s_1 + \beta_{1,2} d^s_2) \tag{14}$$

Putting (14) into (12) and linearizing the equation, the dual-frequency IF combination function model is obtained:

$$\begin{cases} P^S_{r,IF} = \rho^s_r + c(dt_{r,IF} - dt^s_{IF}) + T_{rs} + \varepsilon^S_{r,IF,P} \\ L^S_{r,IF} = \rho^s_r + c(dt_{r,IF} - dt^s_{IF}) + T_{rs} + \lambda_{IF} N^{\hat{s}}_{r,IF} + \varepsilon^S_{r,IF,L} \end{cases} \tag{15}$$

where $dt_{r,IF}$ represents the receiver clock offset to be estimated that absorbs the receiver-side IF code delay. The IF code delay introduced by precision satellite clock products will have to be absorbed by the ambiguity parameters $N^{\hat{s}}_{r,IF}$. They can be expressed as follows:

$$\begin{cases} dt_{r,IF} = dt_r + (\alpha_{1,2} d_{r,1} + \beta_{1,2} d_{r,2}) \\ \hat{N}^s_{r,IF} = N^s_{r,IF} + \left[(\alpha_{1,2} b_{r,1} + \beta_{1,2} b_{r,2}) - (\alpha_{1,2} b^s_1 + \beta_{1,2} b^s_2)\right] / \lambda_{IF} \end{cases} \tag{16}$$

After adopting the ionosphere-free combined model, the parameters to be estimated include the three-dimensional coordinates and receiver clock offset, the tropospheric delay, and the ionospheric-free combined ambiguity of each satellite.

$$X = \left[x, y, z, dt_{r,IF}, T_{rs}, N^s_{r,IF}\right] \tag{17}$$

It is important to note that if the positioning solution is consistent with the dual-frequency observations used to generate the clock offset product, the code delay will be completely eliminated. However, if they are inconsistent, correction through the Differential Code Bias (DCB) corrections is necessary to avoid systematic bias.

In this paper, six-day observation data from seven international GNSS Monitoring and Assessment System (iGMAS) stations (bjf1, chu1, gua1, lha1, sha1, wuh1, and xia1) and two IGS stations (jfng, urum) during DOY 091-096 in 2023 are used to evaluate and analyze the RT PPP positioning accuracy of CNES, PPP-B2b, and BDSBAS RT augmentation information. The distribution of the 9 stations is shown in Figure 6. Note that in Figure 6 the IGS stations are represented by slightly larger blue triangles, while the iGMAS stations are represented by slightly smaller red circles.

BDS-3-only, GPS-only, and GPS/BDS-3 dual-system dual-frequency static PPP experiments were conducted, respectively. The truth coordinates of iGMAS and IGS stations obtained from the snx file were provided by the Product Integration and Service Center (ISC) and IGS, respectively. Since the satellite positions of CNES, PPP-B2b, and BDSBAS are based on satellite antenna phase center positions, there is no need for satellite antenna phase correction. Rather, the IGS analysis center's igs20.atx file should be adopted for receiver antenna phase adjustment. While Timing Group Delay (TGD) adjustments are utilized by BDSBAS from the broadcast ephemeris, DCB corrections are used by CNES and PPP-B2b DCB corrections in augmentation information [31]. The RT PPP software version 1.0 used in this article is self-developed based on GAMP version 1.0 [32] software. More detailed processing strategies about the RT PPP software version 1.0 can be found in Table 2.

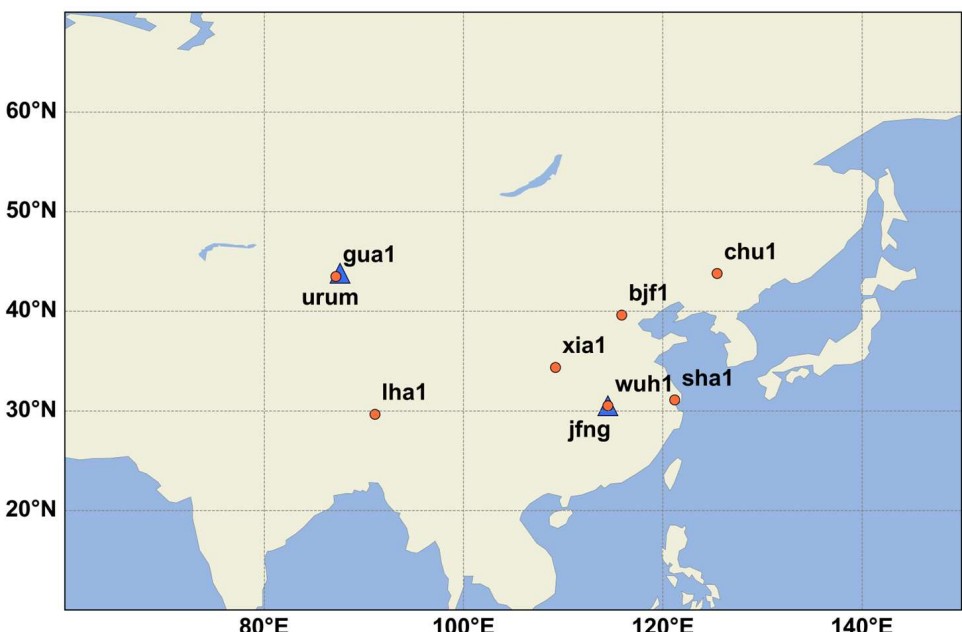

**Figure 6.** Distribution of 9 stations for RT PPP experiments.

**Table 2.** Processing strategies for real-time dual-frequency PPP.

| Project | Processing Strategies |
|---|---|
| Solution type | Dual-frequency static PPP (GPS (L1 + L2)/BDS-3 (B1I + B3I)) |
| Estimator | Kalman filter |
| Observations | Carrier phase and code observation |
| Sample | 30 s |
| Cut-off angle | 10 o |
| Stochastic model | Elevation angle weighting model |
| Ionospheric delay | Ionospheric-free combination |
| Tropospheric delay | Dry component: SAAS + GMF model [29] <br> Wet component: random walk estimation |
| Relativistic effect | Model correction [29] |
| Receiver antenna phase center | Igs20.atx file correction (BDS-3 replaced by PCO of GPS) |
| Phase windup | Model correction [29] |
| Satellite orbit/clock offset | CNES, PPP-B2b, and BDSBAS products |
| Phase ambiguity | Float |

### 3.1.2. RT PPP Experimental Results and Analysis

The 95th percentile value of RT PPP positioning errors is used as an index to express positioning accuracy. Figure 7 shows the positioning accuracy of GPS-only RT PPP for CNES, PPP-B2b, and BDSBAS services over 6 days. It is possible to draw the conclusion that CNES has a positioning accuracy better than 17.5 cm, with a positioning accuracy of 10.8 cm, 2.8 cm, and 8.9 cm in the E, N, and U directions, correspondingly. For PPP-B2b, the positioning accuracy is marginally worse than CNES, with E, N, and U directions' positioning accuracy of 14.3 cm, 3.5 cm, and 11.0 cm, respectively. The positioning accuracy using the BDSBAS service is significantly worse than CNES and PPP-B2b, which in the E, N, and U directions is 69.6 cm, 120.3 cm, and 296.5 cm, correspondingly, and the worst precision in the U direction for the jfng station is 402.0 cm.

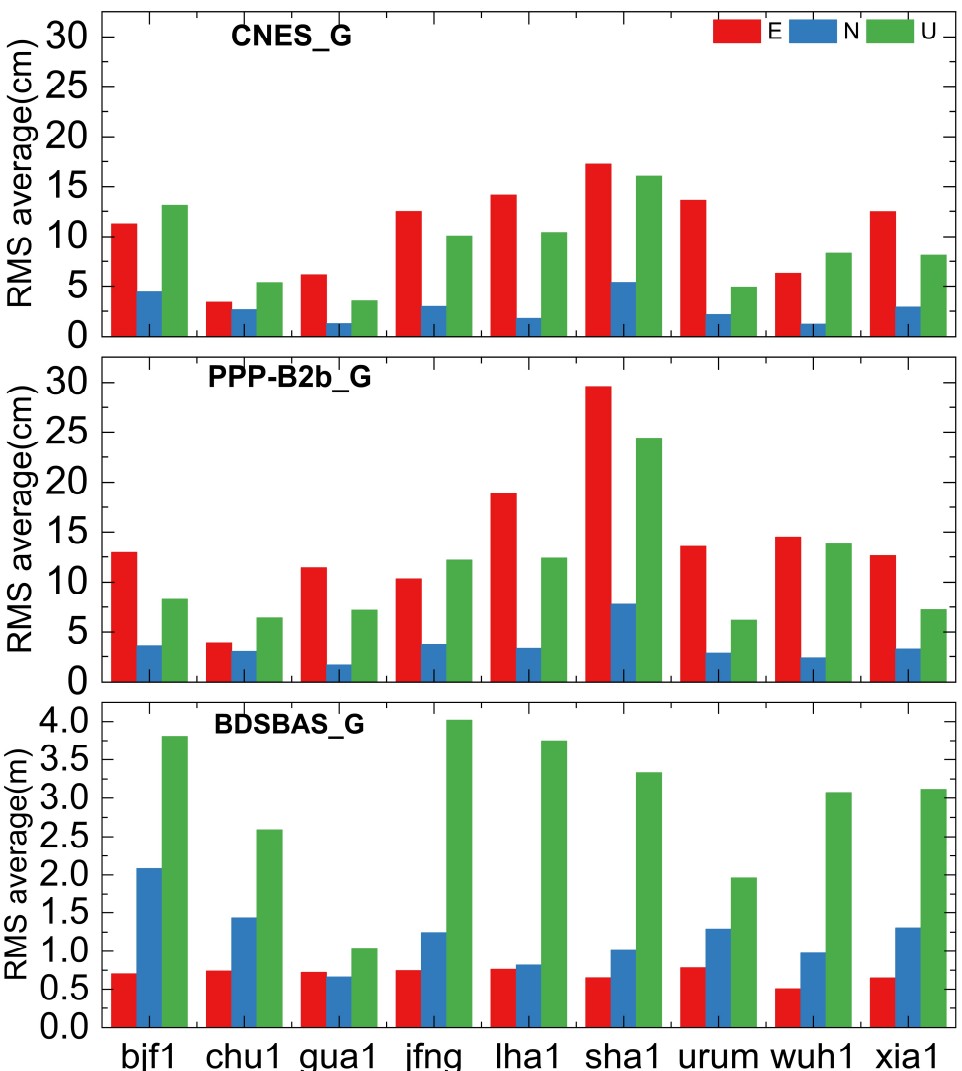

**Figure 7.** Positioning accuracy of GPS based on different augmentation information.

Figure 8 shows the RT PPP positioning accuracy for the BDS-3-only solution based on CNES, PPP-B2b, and BDSBAS RT augmentation information during DOY091-096. For PPP-B2b, the positioning accuracy of the BDS-3-only solution is better than 8.8 cm. In detail, the positioning accuracy, in the E, N, and U directions, is 3.1 cm, 2.4 cm, and 6.2 cm, respectively. The positioning accuracy based on CNES products in all directions is less than 12.4 cm. Specifically, in the E, N, and U directions, it is 4.1 cm, 3.1 cm, and 8.6 cm, respectively. Overall, the positioning accuracy based on PPP-B2b RT augmentation information is marginally superior to that of CNES. In addition, the positioning accuracy based on BDSBAS real-time augmentation information is noticeably worse than that of CNES and PPP-B2b, for the E, N, and U component, which is 22.8 cm, 17.8 cm, and 50.9 cm, correspondingly.

Based on CNES, PPP-B2b, and BDSBAS services, Figure 9 displays the positioning accuracy of the GPS/BDS-3 combined solution during DOY091-096. The RT PPP positioning accuracy based on the CNES RTS is better than 8.6 cm; except for the U direction at the lha1 and sha1 stations, it is within 5.0 cm in all directions. In detail, the positioning accuracy, in the E, N, and U directions, is 3.2 cm, 1.7 cm, and 4.8 cm, respectively. Overall, the GPS/BDS-3 combined positioning accuracy based on the PPP-B2b service is marginally poorer than CNES, with E, N, and U directions' positioning accuracy of 2.8 cm, 2.0 cm, and 5.7 cm, respectively. For BDSBAS, the positioning accuracy, in the E, N, and U directions,

is 21.9 cm, 22.0 cm, and 56.2 cm, correspondingly, which is significantly inferior to that of CNES and PPP-B2b.

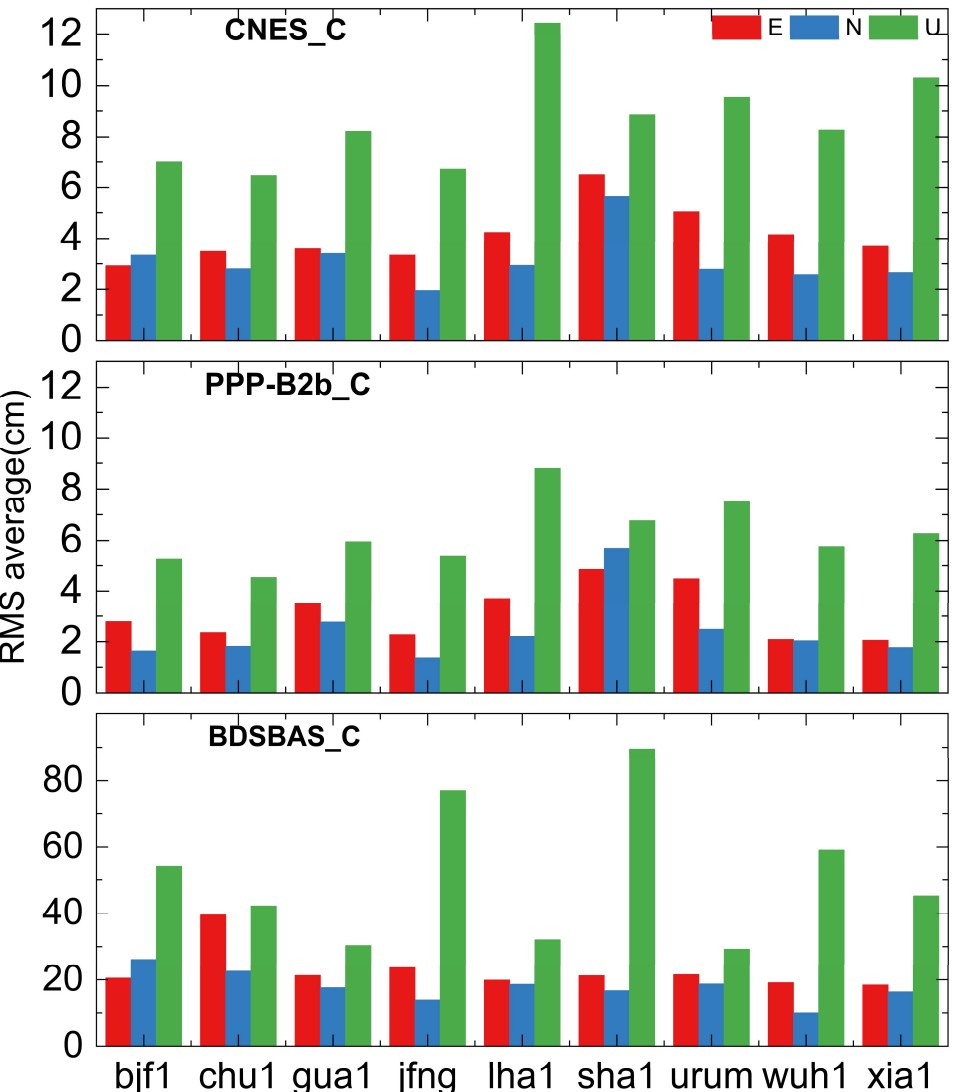

**Figure 8.** Positioning accuracy of BDS-3 based on different augmentation information.

In summary, the PPP performance for the BDS-3-only solution based on the PPP-B2b service is marginally better than CNES, while the GPS-only and GPS/BDS-3 dual systems perform slightly worse than CNES products. The RT PPP accuracy of the BDS-3-only and GPS/BDS-3 dual systems based on BDSBAS RT augmentation information is at the decimeter level, with noticeably better horizontal accuracy than vertical accuracy. The accuracy of a GPS-only system based on BDSBAS services is poor, particularly in the U direction where it can be poorer than 4 m. This is mainly due to the frequent interruption of the RT PPP caused by an insufficient number of correctable GPS satellites that support the L1/L5 frequencies.

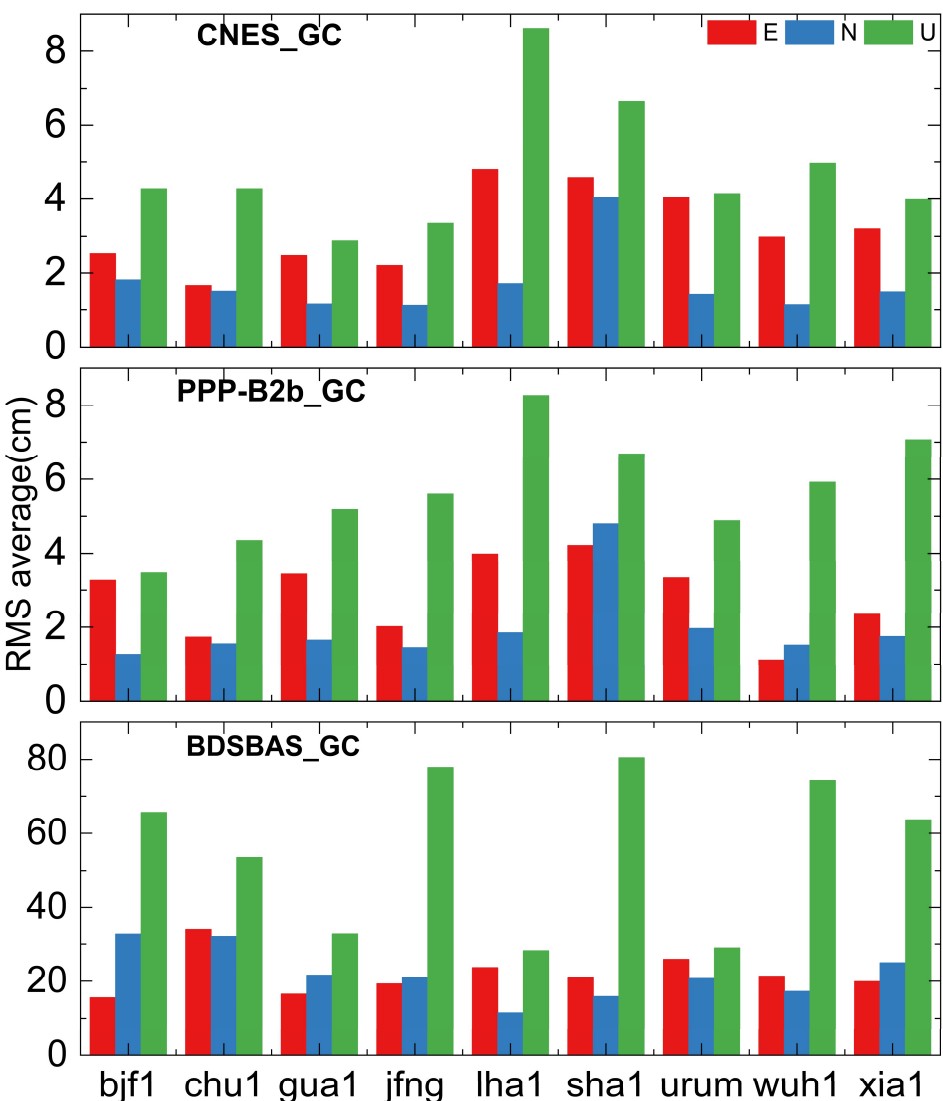

**Figure 9.** Positioning accuracy of GPS/BDS-3 based on different augmentation information.

### 3.2. Positioning Availability Analysis Based on Real-Time Augmentation Information

The empirical availability rate (EAR) is used to evaluate the RT PPP positioning availability of CNES, PPP-B2b, and BDSBAS real-time products, which is determined if the epoch can be used for RT PPP [33,34]. The EAR formula is as follows:

$$\omega = \frac{epoch_{\text{Available}}}{epoch_{all}} \tag{18}$$

In this paper, the positioning results of seven iGMAS stations, i.e., bjf1, chu1, gua1, lha1, sha1, wuh1, and xia1, and two IGS stations: jfng and urum for 6 consecutive days from DOY091 to 096 in 2023 are evaluated to analyze the positioning availability of three services. The details are shown in Figure 10. Except for the xia1 station, the positioning availability of the GPS/BDS-3 combined solution based on the CNES, PPP-B2b, and BDSBAS service is greater than 99.9%. The positioning availability of the GPS/BDS-3 combined and BDS-3-only solution based on CNES and PPP-B2b real-time augmentation information is similar, which is beyond 99%. The availability of the GPS-only solution based on PPP-B2b and BDSBAS is poorer than that of BDS-3-only. The availability of the GPS-only solution based on PPP-B2b is approximately 90%, while that of BDSBSA is around 25%. This is due to the frequent interruption of the RT PPP caused by an insufficient number of correctable GPS satellites of BDSBAS that support the L1/L5 frequencies. The positioning availability of the

GPS-only, BDS-3-only, and GPS/BDS-3 combined solution of CNES real-time products is very similar. For PPP-B2b and BDSBAS, the GPS/BDS-3 combined solution is superior to the BDS-3-only solution and GPS-only solution.

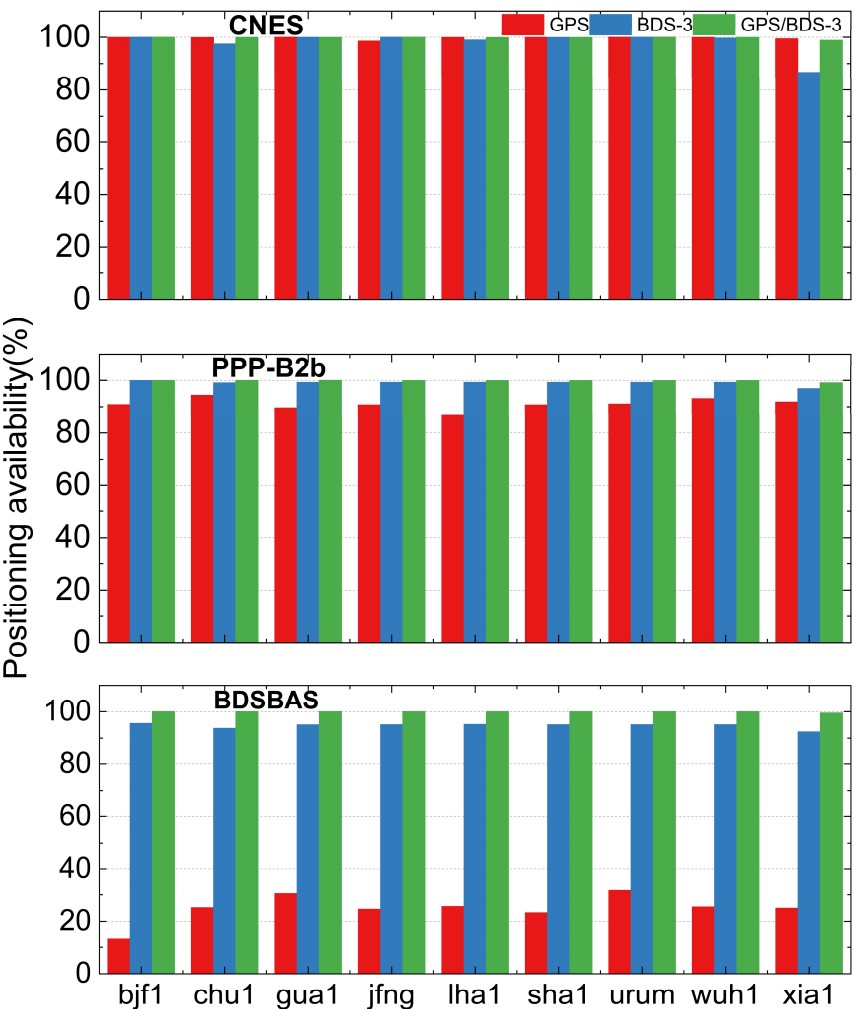

**Figure 10.** Positioning availability based on different augmentation information.

## 4. Application Field Analysis

Table 3 presents the details of CNES, PPP-B2b, and BDSBAS in terms of broadcasting modes, service areas, tracking station networks, and space-time references. As most industries lack unified standards for user terminal navigation and positioning requirements, this paper refers to relevant resources and statistics to outline the user terminal navigation and positioning requirements of major industries.

**Table 3.** The basic information of CNES, PPP-B2b, and BDSBAS [3–6].

| Augmentation Information Source | Broadcast Method | Service Area | Tracking Station Network | Spatial-Temporal Reference |
|---|---|---|---|---|
| CNES RTS | Internet | Global | Global | WGS84/GPST |
| PPP-B2b | GEO satellites | China and surrounding regions | Regional + inter-satellite links | BDCS/BDT |
| BDSBAS | GEO satellites | China and surrounding regions | Regional | BDCS/BDT |

Based on the relevant literature [35–39], this article presents a summary of user requirements for navigation and precise positioning in various important domains, as shown in Table 4. By considering the information from Table 3, Table 4, and the analysis of RT PPP performance and availability discussed in Section 3, it is observed that the CNES real-time product is mainly applicable in industries with global network coverage, such as environmental monitoring and inspection applications in the environmental protection industry, agricultural automatic driving applications, railway vehicle regulation and automatic operation and maintenance inspection applications, highway operation and maintenance inspection and vehicle management applications, municipal bus dispatching and management applications, and cultural tourism tourist guidance; PPP-B2b real-time augmentation information is mainly applicable to China and surrounding areas, especially internet-restricted areas, and the specific application industry is generally the same as CNES real-time augmentation information. However, since it is not restricted by the internet, it is also applicable to emergency search and rescue after disasters, navigation in deserts and mountains, etc. For BDSBAS real-time augmentation information, its enhanced positioning accuracy is poor, but the integrity and continuity not listed in this paper have significant advantages, which are mainly used in aviation and navigation fields.

**Table 4.** The main industry user-side navigation and positioning requirements [35–39].

| Category | Application | Accuracy (RMS) | Coverage Area | Availability |
|---|---|---|---|---|
| Environmental Protection | Environmental Monitoring, Inspection | 0.5–2 m | Protected Areas | 95% |
| Agriculture | Agricultural Machinery Autonomous Driving | 0.1 m | Cultivated Areas | 99% |
| Railways | Vehicle Control, Automatic Maintenance Inspection | 0.1 m | Railway Lines | 99.5% |
| Highways | Maintenance Inspection, Vehicle Management | 0.1 m | Highways | 99.5% |
| Emergency Rescue | Emergency Communication | 0.1 m | Production Areas | 98% |
| Municipal | Bus Dispatch and Management | 0.5 m | Urban Areas | 98% |
| Tourism | Tourist Guidance | 1.0 m | Scenic Areas | 95% |
| Surveying and Mapping | Surveying, Engineering Measurement | 0.02 m | Entire Area | 98% |
| Aviation | Aircraft Landing Guidance, Route Planning | 4 m | Airport Areas | 99% |
| Maritime | Ship Entry and Exit Guidance, Route Planning | 10 m | Port Areas | 99% |

## 5. Conclusions

In this paper, several methods of real-time augmentation information acquisition and the RT orbit and RT clock offset recovery available in China are introduced, and the performance of CNES, PPP-B2b, and BDSBAS augmentation information is evaluated in terms of real-time orbit and clock offset accuracy, positioning accuracy, and positioning availability.

(1) PPP-B2b and CNES RT orbit accuracy is consistent in the radial direction, PPP-B2b GPS RT orbit accuracy is marginally poorer than CNES, and BDS-3 orbit accuracy is slightly better than CNES but within 5 cm. PPP-B2b RT orbit accuracy is worse than CNES in the A and C direction, both are at the decimeter level. For CNES, except IGSO satellites, the orbit accuracy of other satellites is about centimeter-level. BDSBAS real-time orbit accuracy is worse than CNES and PPP-B2b, which reached the decimeter level.

(2) The performance of the CNES and PPP-B2b RT clock offset are consistent, the CNES GPS RT clock offset accuracy is marginally higher than that of PPP-B2b, and the BDS-3 real-time clock offset accuracy is marginally inferior to that of PPP-B2b, especially for the IGSO satellite. The BDSBAS RT clock offset accuracy is considerably inferior to that of CNES and PPP-B2b, but it has reached the sub-nanosecond level.

(3) The RT PPP performance of BDS-3-only based on the PPP-B2b service is marginally greater than the CNES service and that of GPS-only and GPS/BDS-3 dual systems are marginally worse than CNES, but it still reaches centimeter level. The RT PPP performance of the BDSBAS service is at the decimeter level, except for the U component of GPS-only, which may be caused by the limited quantity of GPS satellites that can be corrected.

(4) The RT PPP positioning accuracy and availability of GPS/BDS-3 and BDS-3-only are on par with CNES and PPP-B2b services. The acquisition of RT augmentation information is where the two services' applications diverge most. The CNES RT service relies on an unobstructed network environment, while PPP-B2b only serves China and surrounding areas. For BDSBAS RT services, its positioning integrity and continuity have significant advantages, mainly applied in aviation and maritime fields.

**Author Contributions:** Conceptualization, M.W., L.W. and B.C.; Methodology, L.W.; Software, M.W. and F.Y.; Validation, M.W., W.X., F.Y. and B.C.; Investigation, W.X.; Resources, L.W.; Writing—original draft, M.W.; Writing—review & editing, M.W., L.W., W.X., F.Y. and B.C.; Visualization, M.W.; Supervision, L.W. and W.X.; Project administration, L.W.; Funding acquisition, L.W. All authors have read and agreed to the published version of the manuscript.

**Funding:** This research was funded by the provincial Key R&D Program of Shaanxi (2022KW-09), the Key Research and Development Program of Shaanxi Province in 2021 (2021LLRH-06-03-06) and the APC was funded by [Le Wang].

**Data Availability Statement:** The data presented in this study are available on request from the corresponding author. The data are not publicly available due to privacy reasons.

**Conflicts of Interest:** The authors declare no conflict of interest.

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
