# Peer review of "Performance Evaluation and Application Field Analysis of Precise Point Positioning Based on Different Real-Time Augmentation Information"

_remotesensing, doi:10.3390/rs16081349_

Round 1

Reviewer 1 Report (Previous Reviewer 3)

Comments and Suggestions for Authors

1. The author points out that “Users can access PPP-B2b real-time services by using special receivers through B2b signals”. Is the real-time enhancement information obtained by special receiver accurate?

2. It is shown that the GPS satellite orbit accuracy in R direction from CNES, which is better than 3.5 cm superior than the accuracy in the C and A component. What is the accuracy of R, A and C components respectively?

3. The author uses STD as an indicator of real-time clock offset accuracy analysis. Please explain why the mean is not used.

4. The paragraph following Formula ( 1 ) should not be indented first.

5. There are some grammatical errors in the paper, for example, in line 217, “do not support” is wrongly written as “are not support”.

Author Response

Reviewer 2 Report (New Reviewer)

Comments and Suggestions for Authors

The authors compare PPP performances using IGS-CNES, BDS B2b and BDSSBAS corrections. Some parts of the text are in red, which I guess means that this is not the first version of the manuscript. This version is of good quality, and I only have minor comments.

In the introduction, before line 62, it would help to explain in a few words that BDS B2b provides corrections for BeiDou-3 and GPS satellites.  No all readers are familiar with that.

Line 82: the term “ocean motion positioning” is unclear to me.  Is it the positioning of a ship?

Line 90: “signal frequency (SF)” should be “single frequency (SF)”

Line 121: BDSSBAS is not transmitted on B1C/B2a, but on dedicated SBAS L1 and L5 signals.  To be corrected.

Line 322: “Therefore, the satellite clock offset products differs from the precision.”.  This sentence is unclear to me. To be clarified what “the precision” means.

Eq (15): in the first equation, shouldn’t it be (d_tr_IF -dt^S_IF)  (IF subscript missing)?

Eq (15): the phase bias terms (b_r and b_s) of (12) seem to have disappeared.  To be corrected or explained.

Comments on the Quality of English Language

The English usage could be improved.  Some sentences are malformed, sometimes affecting the readability.

Two examples among many others:

line 149: "Real‐time orbit recovery for PPP‐B2b service, multiple IOD should be correctly matched" --> "For real-time orbit..."

line 157: "the difference is that the calculation method of the long semi‐axis and satellite motion speed" --> "that" should be deleted.

Author Response

Reviewer 3 Report (New Reviewer)

Comments and Suggestions for Authors

The article deals with an important problem of the application of RT PPP, which has a significant practical aspect. The article can be accepted, but some points need to be clarified or corrected.

1. The title of the article has a somewhat generalized character, which does not allow to reveal its specificity and does not indicate the novelty of the research.

2. As far as I know, BDSBAS and PPP-B2b are two types of embedded satellite-based augmentation services of BDS-3, but with different accuracy characteristics. Can we consider them as equivalent systems to compare them?

3. Since RT augmentation information from CNES is transmitted through a ground caster, it is known a priori that its quality will be better than the corresponding information transmitted through geostationary satellites. What then is the point of the comparison?

4. Satellite position prediction requires accurate dynamic models, but uncertainties in the models can lead to extrapolation errors, especially over long periods of time, which is the case when augmentation information from GEO satellites is used. If in statistical terms everything can be more or less good (see fig. 1 and fig. 2), then in real time everything will be very unfortunate. Did the authors ask themselves the same question?

5. The procedures for restoring orbits and correcting real-time clocks described in section 2 require specifics of the practical plan, namely, how it was done: with the help of some known software or your own?

6. Subsection 3.1.1 (RT PPP theory and processing strategy) is not entirely clear. What emerges from it and what is used in further research? This especially applies to formulas (10)-(17).

7. In formula (10), the correction for the troposphere should probably be denoted by , like the rest of the components.

8. In subsection 3.1.2. "RT PPP experimental results and analysis" does not indicate how the strategy presented in Table 2 was implemented. The same question comes up again, what software was used?

9. Statistical estimates are given in the previously mentioned subsection (see fig. 7-9 and fig. 10). All this can be true only in post-processing, and in relation to the real-time mode, such generalized estimates do not fully characterize the process. Can it be shown in any other way?

10. What new did the authors get, in addition to what is already known, in the field of application of RT PPP.

Author Response

This manuscript is a resubmission of an earlier submission. The following is a list of the peer review reports and author responses from that submission.

Round 1

Reviewer 1 Report

Comments and Suggestions for Authors

Except for the one below point, I found no other issue that should be corrected. 

1-) line 197-198, the authors stated that 'The accuracy of A and C component is marginally poorer than R component for PPP-B2b'

Since the along-track components suffer strongly from error accumulation, this shouldn"t be the case (the accuracy of A and C shouln't be poorer than R component). 

To purely deconjugate the RAC components as much as possible, we should work on the inertial system (ECI). 

I recommend authors recompute the RAC components in the ECI frame (at least one data set), and check whether this inconsistency still continues.

The authors can use this website https://hpiers.obspm.fr/eop-pc/index.php?index=matrice  for ECEF>>ECI conversion. 

If this inconsistency is gone after the conversion, all RAC values should be computed in the ECI frame.

Reviewer 2 Report

Comments and Suggestions for Authors

Review of the manuscript entitled "Performance evaluation of precise point positioning based on multiple real‐time augmentation information".

The manuscript submitted for review concerns comparison among CNES, PPP-B2b and BDSBAS real‐time augmentation information. As a justification for taking up this topic, the authors delare the lack of a systematic comparison between BDSBAS, PPP-B2b and CNES. Unfortunately, both the stated aim and its justification and implementation are minimally scientific in nature. The manuscript is a technical report rather than a scientific article and therefore has no scientific soundness. Moreover, right from the beginning, you can expect the overall results of the comparison. Unfortunately, the manuscript does not provide the methodology or a full description of the data used. The comparison is also limited to a regional, or even local, range and is carried out only for the period of 6 days and for 6 stations. The conclusions are only a repetition of selected comparison results. There is a lack of discussion at the manuscript.

References, although from international journals, do not represent research from recognized scientific centers around the world. The references provision is inconsistent and, in some cases, inconsistent with the journal's guidelines.

There are also minor errors in the manuscript, such as: e.g. linguistic errors (lack of predicates in sentences, some sentences begin with lowercase letters, e.g. title); in the abstract the abbreviation CNES is introduced without prior explanation, reference number [25] appears on line 40; in the formulas there are the symbols r,a,c, while in the text R, A, C; the charts are of poor quality and the y-axis range is different for different services; key words are too general; no source of information included in table 4.

Reviewer 3 Report

Comments and Suggestions for Authors

1. The statement in the abstract that “the research on the performance evaluation, comparison and application scope of these three products is still unknown and unrevealed” is too absolute and should be revised.

2. According to the authors, “The BDS‐3/GPS kinematic RT PPP positioning also arrived centimeter‐level placement, its availability exceeds 70% in Asia, and over 80% in China.” Is this percentage data available for experimental analysis? If not, please indicate the source of the data in this article.

3. In section 2.3.1, BDSBAS can only use information data from 18 GPS satellites for performance evaluation, which is less compared to PPP-B2b and CNES.

4. In section 2.3.2, the author replaced the C39 satellite data with the C40 satellite data that should have been used, and please provide additional information on the feasibility and implications of that replacement.

5. In Figure 6, the location of 6 sites should be indicated according to the illustration, of which 2 sites are coincident in the figure, but the coincident site drawing is not very clear. It is recommended to modify figure 6 and re-label the sites.
